# GenUDC: High Quality 3D Mesh Generation With Unsigned Dual Contouring Representation

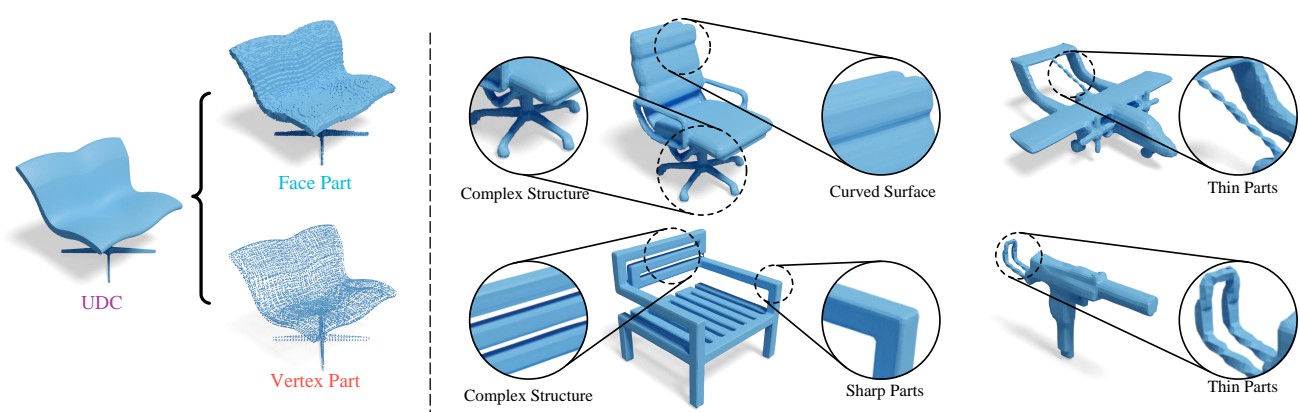

(a) Unsigned Dual Contouring Representation       (b) Mesh generation samples of GenUDC

**Figure 1: (a) A visual sample of Unsigned Dual Contouring Representation (UDC) consisting of the face part and vertex part. (b) Our high-quality mesh generation results in $64^3$ resolution with close-up views.**

## ABSTRACT

Generating high-quality meshes with complex structures and realistic surfaces is the primary goal of 3D generative models. Existing methods typically employ sequence data or deformable tetrahedral grids for mesh generation. However, sequence-based methods have difficulty producing complex structures with many faces due to memory limits. The deformable tetrahedral grid-based method MeshDiffusion fails to recover realistic surfaces due to the inherent ambiguity in deformable grids. We propose the GenUDC framework to address these challenges by leveraging the Unsigned Dual Contouring (UDC) as the mesh representation. UDC discretizes a mesh in a regular grid and divides it into the face and vertex parts, recovering both complex structures and fine details. As a result, the one-to-one mapping between UDC and mesh resolves the ambiguity problem. In addition, GenUDC adopts a two-stage, coarse-to-fine generative process for 3D mesh generation. It first generates the face part as a rough shape and then the vertex part to craft a detailed shape. Extensive evaluations demonstrate the superiority of UDC as a mesh representation and the favorable performance of GenUDC in mesh generation. The code and trained models will be released upon publication.

**Unpublished working draft. Not for distribution.**

## CCS CONCEPTS

• **Computing methodologies** → **Mesh models**; • **Information systems** → *Multimedia content creation*.

## KEYWORDS

Mesh, 3D Generation, Diffusion Model, Dual Contouring

## 1 INTRODUCTION

Mesh plays an important role in 3D content generation and reconstruction [21, 40, 41, 68], AR/VR [22], robotics [20, 65], and autonomous driving [4, 9, 18] and other 3D tasks. It can flexibly represent various complex geometric shapes. High editability allows meshes to be modified and adjusted easily in computer-aided design (CAD). Additionally, it is effortless for users to convert meshes to other 3D representations, e.g., voxels, point clouds, and neural implicit functions. Besides, the rendering pipelines are designed for meshes, enabling high-quality 3D visualization effects. However, Employing deep neural networks on meshes is tricky because the numbers of vertices and faces are constantly changing, and modeling the complex topology structure of faces is also an obstacle. To navigate those challenges, a mesh representation compatible with deep learning and a capable generative framework adapted to this mesh representation are both highly desired.

Most existing approaches focus on intermediate representations, e.g., voxels [57, 71], point clouds [1, 45, 46], neural implicit functions [12, 33, 68, 77, 78] and so on, which are highly compatible with deep learning. However, those methods require a post-processing step [10, 13, 15, 28, 43, 58] to extract meshes, resulting in over-smooth surfaces and lacking detailed geometry. PolyGen [50] first treats vertices and faces as sequences and uses transformer networks

[66] to generate vertices and then faces. MeshGPT [63] and Poly-Diff [2] follow similar ideas but concentrate on faces. All three approaches cannot produce mesh with intricate geometry since the memory limits the number of faces to no more than 2800. MeshDiffusion [42] chooses to combine a deformable tetrahedral grid with Signed Distance Functions (SDF) to model meshes. However, its data preparation is especially slow (Tab. 6), and the generated meshes are crumpled due to the deformable nature of the grid and the inaccurate 2D supervision.

To address these two challenges, we construct a novel framework dubbed **GenUDC** to combine the Unsigned Dual Contouring representation (UDC) with a two-stage, coarse-to-fine generative process, enabling high-quality mesh generation. As a mesh counterpart, UDC consists of a face part and a vertex part. Accordingly, we decompose the mesh generation into two subtasks: the face part generation and vertex part generation. To address these subtasks, we devise the two-stage, coarse-to-fine pipeline, which involves generating faces first and then vertices.

Precisely, to find a proper mesh representation, we expand Dual Contouring [28], which has long been regarded as an isosurface reconstruction method, to generation tasks. Thus, we obtain the UDC representation to model meshes as shown in Fig. 1 (a) and Fig. 2. In UDC, we discretize a mesh in a regular grid. The faces part of UDC is a set of tiny faces represented by boolean values. The vertex part of UDC contains all the actual and potential vertices of those tiny faces. Since the values of the face part and vertex part are arranged in a regular grid, we can conveniently employ deep learning-based generative models to learn the distribution of UDC.

Another pivotal component is the two-stage, coarse-to-fine generative process specially designed for UDC. Because the mesh is discretized in UDC, the face part draws the rough shape, and the vertex part describes the details. Consequently, we first employ a latent diffusion model to generate the face part, determining the mesh's rough shape and topological structure. Then, conditioning the rough shape, we take a vertex refiner to generate the vertex part. Such a two-stage, coarse-to-fine pipeline is a natural solution. Without this pipeline, the edges would be jagged due to the inaccurate vertex part. We will study the necessity of this two-stage, coarse-to-fine pipeline in Sec. 4.4.

Finally, using GenUDC, we can produce high-quality meshes with complex structures and realistic details as shown in Fig. 1 (b) and Fig. 3. Comprehensive experiments demonstrate the superiority of our method over existing ones in mesh generation. In data fitting, compared with MeshDiffusion, our method runs at 3274% times their speed and consumes only about 13% of their total memory as shown in Tab. 6.

To summarize, the contributions of this paper are:

- We propose a novel generative framework, called **GenUDC**, utilizing UDC as the representation for high-quality mesh generation.
- We design a two-stage, coarse-to-fine generative pipeline to UDC, which generates faces and then vertices, circumventing the jagged edges problem.
- Extensive experiments demonstrate our method's superior performance in mesh generation and data fitting.

**Table 1: Taxonomy of mesh generation methods.**

| Method | Representation | Memory | Maximum Num Of Faces |
|---|---|---|---|
| PolyGen [50] | Face Sequence + Vertex Sequence | High | Less Than 2800 |
| MeshGPT [63] | Triangle Face Sequence | High | Less Than 800 |
| PolyDiff [2] | Triangle Face Soup | High | Less Than 800 |
| MeshDiffusion [42] | Deformable Tetrahedral Grid + SDF | Medium | More Than 32768 |
| **Ours (GenUDC)** | **UDC (Regular Grid)** | **Medium** | **More Than 32768** |

## 2 RELATED WORK

In this section, we will outline some closely related topics to our study: 3D shape generation, isosurface reconstruction, and diffusion models.

### 2.1 3D Shape Generation

With the advent of deep learning, researchers have been exploring the generation of 3D voxels [34, 57, 67, 71, 73] and point clouds [1, 3, 17, 26, 29, 32, 36, 45, 46, 61, 72, 74] using neural networks. However, voxels suffer from memory limits, and point clouds lack topology of shapes. Until the dawn of neural implicit function [12, 49, 53], the community finds it an excellent shape representation, which does not require a lot of memories and can be easily transformed into meshes. The neural implicit function is specially designed for advanced deep neural networks and inspires a lot of work [14, 16, 25, 27, 31, 33, 35, 37, 48, 51, 60, 62, 68, 70, 76–78]. It utilizes SDF values or occupancy values as the intermediate representation of 3D shapes. By using some isosurface reconstruction methods like Marching Cube [15, 43], and Dual Contouring [28], meshes can be reconstructed from those neural implicit functions. However, this also means those implicit function-based methods still require a post-processing step and cannot directly generate meshes.

A collection of existing methods [6–8, 11, 39, 54, 64, 69, 75] adapt Neural Radiance Fields (NeRF) or 3D Gaussian Splatting (3DGS) as 3D representation. They utilize the powerful text-to-image generative model, Stable Diffusion [55], as the guidance to optimize NeRF or 3DGS with a text prompt. After optimization, the final NeRF or 3DGS contains both 3D shape and texture information. The mesh can be extracted from it by some post-processing methods. However, they are time-consuming, taking hours of optimization for each text prompt. They also suffer from artifacts such as over-saturated colors and the multi-face problem.

Moreover, some works are trying to find a proper mesh representation to generate meshes directly. PolyGen [50], MeshGPT [63], and PolyDiff [2] are inspired by natural language processing to process meshes as sequences. By leveraging the power of transformer network [66], they can theoretically produce vertices and faces of any length. In practice, the limited memory constrains the complexity of synthetic mesh structures, making it difficult for them to generate curved surfaces. In MeshDiffusion [42], a deformable tetrahedral grid and SDF values are utilized to recover meshes. It supposes all mesh vertices are on the edges of the deformable tetrahedral grid. It can use linear interpolation to compute mesh vertices with the coordinates of adjacent grid points and SDF values. After getting the mesh vertices, It produces faces by connecting mesh

vertices in the same tetrahedrons. However, the deformable grid brings ambiguity to the fitting mesh. Two different tetrahedral grids with distinct SDF values may recover the same mesh. In addition, to fit a mesh, the deformable grid is trained in the supervision of rendered images, which are inaccurate due to various rendering settings. The ambiguity and 2D supervision tend to result in deficient surfaces shown in Fig. 7. As for data preparation, it takes too much time and memory to fit a tetrahedral grid on a shape due to the 2D supervision, as shown in Tab. 6.

In contrast to sequence-based methods, e.g., PolyGen, MeshGPT, and PolyDiff, our method is capable of using limited memory to generate a diverse range of mesh structures, such as flat surfaces, thin parts, curved surfaces, sharp parts, and so on, as shown in Fig. 1 (b). Compared with MeshDiffusion, we use a regular grid to fit meshes with more accurate results, less processing time, and less memory, as shown in Tab. 6 and Fig. 7. We taxonomize methods that can directly generate meshes in Tab. 1. We present more details of the data fitting comparison between MeshDiffusion and ours in Sec. 4.5.

## 2.2 Isosurface Reconstruction

Typically, isosurface reconstruction methods extract meshes from volume data (e.g. voxels and SDF volumes). As a pioneering work, the original Marching Cubes (MC) method is proposed by Lorensen and Cline [43]. It discretizes a mesh into a regular grid and creates approximative surfaces in each cube according to intersections between the mesh and grid. Its most well-known variant, MC33 [15], can even model all possible topological cases in a cube. However, since vertices of approximative surfaces are on the edges of the grid, it is hard for the marching cubes method to model sharp parts. The Dual Contouring method (DC) [28] is thus proposed. Its vertices of approximative faces (also called dual faces) are in the cubes. So DC can recover sharp parts. With the rise of deep learning, Deep Marching Cubes [38] first applied deep learning to isosurface reconstruction. Neural Marching Cubes (NMC) [13] and Neural Dual Contouring (NDC) [10] focus on building a learnable version of MC and DC. Manifold Dual Contouring [58] and FlexiCubes [59] try to solve the non-manifold problem in DC. VoroMesh [47] introduces Voronoi diagrams to isosurface reconstruction.

All isosurface reconstruction methods focus on developing a pipeline to transform a kind of 3D data into a counterpart of mesh. In contrast, we adopt UDC and expand it to shape generation by learning the distribution of the UDC representations. In other sections, with a little abuse of the abbreviation, we refer to UDC representation as UDC.

## 2.3 Diffusion Models

Diffusion models are a class of deep generative models that play an important role in artificial intelligence generated content (AIGC). It achieves pleasant results in various applications, such as image generation [24, 55], image super-resolution [55], shape generation [42], text-to-3D [16, 35], etc. Diffusion models are designed to model the step-by-step transformations between a simple distribution (e.g. Gaussian distribution) and a complex distribution of data. Once trained, a diffusion model can map a sample of the simple distribution to the desired data distribution. As a milestone of diffusion

models, the Denoising Diffusion Probabilistic Model (DDPM) [24] introduces variational inference into diffusion models and shows greater potential over generative adversarial networks [19]. But it still suffers from the huge memory requirement. Therefore, the latent diffusion model (LDM) [55] proposes to train diffusion models in a low-dimensional latent space instead of the high-dimensional data space. It has been demonstrated that this technique can speed up training and reduce memory footprints without degradation of generation quality. In this paper, we adopt the LDM in the face part generation (see Sec. 3.3) since the regular grid takes a lot of memory footprints.

## 3 METHOD

### 3.1 Overview

Aiming at mesh generation, how to represent meshes, and how to process meshes with neural networks are two critical issues. To address them, we propose GenUDC, a novel generative framework for mesh generation. In GenUDC, we discretize a mesh in a regular grid to get its corresponding Unsigned Dual Contouring representation (UDC). Thus, due to the regular grid structure of UDC, neural networks can easily be used on both watertight and non-watertight meshes. We further propose a two-stage, coarse-to-fine pipeline adapted to UDC, which generates faces and vertices successively. In summary, we offer a new and straightforward solution for mesh generation.

In the following sections, we first elaborate on UDC in Sec. 3.2. Then, we illustrate our generative models for face generation in Sec. 3.3 and vertex generation in Sec. 3.4. Finally, the implementation details are presented in Sec. 3.5.

### 3.2 Unsigned Dual Contouring Representation

We have briefly shown the main ideas of the Unsigned Dual Contouring representation (UDC) in Fig. 1 (a) and Fig. 2. For more details, in a grid $\mathcal{G}$ with the size of $(X+1, Y+1, Z+1)$, UDC can be formalized as:

$$\text{UDC} = \begin{cases} \mathcal{V} \in \mathbb{R}^{3 \times |C|}, & \text{(vertex part)} \\ \mathcal{F} \in \mathbb{B}^{|\mathcal{E}|}, & \text{(face part)} \end{cases} \tag{1}$$

where $C$ are the cubes in the grid, $\mathcal{V}$ are the vertices, $\mathcal{E}$ are the edges inside the grid, and $\mathcal{F}$ are the faces (also called dual faces) denoted by the intersection flags of edges. The grid $\mathcal{G}$ contains $(X+1)(Y+1)(Z+1)$ nodes. There are $|C| = XYZ$ cubes in the grid, and each cube contains a vertex $v \in \mathcal{V}$. Considering the edges along the x-axis, y-axis, and z-axis, there are $|\mathcal{E}| = X(Y-1)(Z-1) + (X-1)Y(Z-1) + (X-1)(Y-1)Z$ inside edges. If the intersection flag of an edge is true, four adjacent vertices make up two triangle faces that are *dual* to the edge. In other words, the edge intersects with one of the two triangle faces when the flag is true. If not, there is no face intersecting with this edge. When translating a UDC to a correlative mesh, we craft faces by traversing all intersection flags in $\mathcal{F}$ and remove a subset of $\mathcal{V}$ which are not in these faces. By this means, faces and remaining vertices comprise the final mesh.

Compared with the traditional SDF-based methods [42, 77], which usually generate over-smooth shapes, UDC can easily model the sharp parts as shown in Fig. 7. In addition, the rigid grid used in UDC is suited for deep neural networks and can produce more

**Figure 2: The overview of GenUDC. It consists of UDC and a two-stage, coarse-to-fine generative pipeline. We first translate the meshes to UDCs by data fitting. Then, we take UDCs to train the generative models. After training, we can generate the face part and vertex part to compose the output UDC.**

realistic surfaces than the deformable grid of MeshDiffusion [42], which will be evaluated in Sec. 4.5. Moreover, UDC has the potential to model non-watertight shapes shown in Fig. 7.

In practice, $\mathcal{V}$ are the relative coordinates in each cube, which means $0 \leq min(\mathcal{V})$ and $max(\mathcal{V}) \leq 1$ and $\mathcal{F}$ are boolean values. When $X = Y = Z$, we pad $\mathcal{F}$ with zeros to the same size as $\mathcal{V}$. We call $\mathcal{V}$ as the vertex part and $\mathcal{F}$ as the face part.

*Data Fitting.* We follow a similar procedure of DC [28] to fit a mesh with UDC. Given a mesh $\mathcal{M} = (\mathcal{V}^{\mathcal{M}}, \mathcal{F}^{\mathcal{M}})$ and a grid $\mathcal{G} = (C, \mathcal{E})$, we first find the crossing vertices $\mathcal{V}^{\mathcal{E}}$ of the mesh $\mathcal{M}$ on the edges $\mathcal{E}$. Then, we compute the normals $\mathcal{N}^{\mathcal{E}}$ of $\mathcal{M}$ at those crossing vertices. With $\mathcal{V}^{\mathcal{E}}$ and $\mathcal{N}^{\mathcal{E}}$, we can create UDC as:

$$\mathcal{V} = f_{\mathcal{V}}(\mathcal{V}^{\mathcal{E}}, \mathcal{N}^{\mathcal{E}}), \quad (2)$$

$$\mathcal{F} = f_{\mathcal{F}}(\mathcal{V}^{\mathcal{E}}, \mathcal{E}). \quad (3)$$

The dual contouring vertices $\mathcal{V}$ should be on the surfaces of $\mathcal{M}$. So we extrapolate neighboring normals $\mathcal{N}^{\mathcal{E}}$ to find a point of best fit in each cube:

$$f_{\mathcal{V}} : \{v_{xyz} | \arg\min_{v_{xyz}} \sum_{e \in C_{xyz}} (\mathcal{N}^{\mathcal{E}}_e \cdot (v_{xyz} - \mathcal{V}^{\mathcal{E}}_e))^2\}, \quad (4)$$

where $v_{xyz}$ is the vertex inside the cube $C_{xyz}$ which is indexed by $(x, y, z)$, and $e$ are 12 edges of $C_{xyz}$. $0 \leq x < X$, $0 \leq y < Y$, and $0 \leq z < Z$. By default, if there is no $\mathcal{V}^{\mathcal{E}}_e$ or $\mathcal{N}^{\mathcal{E}}_e$ in a cube, $v_{xyz}$ is set to $[0.5, 0.5, 0.5]$.

Besides, we only craft faces $\mathcal{F}$ when $\mathcal{M}$ intersects with an edge $e \in \mathcal{E}$ at the crossing vertex $v \in \mathcal{V}^{\mathcal{E}}$:

$$f_{\mathcal{F}} : \begin{cases} 1, & \text{if } \forall e \in \mathcal{E}, \exists v \in \mathcal{V}^{\mathcal{E}} \text{ is on the } e, \\ 0, & \text{otherwise.} \end{cases} \quad (5)$$

### 3.3 Face Part Generation

In UDC, we have devised a simple and intuitive method for generating faces by connecting them with intersection flags. If an edge's intersection flag is true, it crosses faces. If not, there is no face. By this means, we can denote all faces of the mesh as boolean values and arrange them into a regular grid as a face tensor $\mathcal{F} \in \mathbb{B}^{|\mathcal{E}|}$. We can easily employ typical neural networks to face part generation thanks to the regular grid.

To reduce the memory footprint, we use a Latent Diffusion Model (LDM) [55] to learn the distribution of $\mathcal{F}$. Our LDM consists of a Variational AutoEncoder (VAE) [30] and a diffusion model [24, 55]. VAE compresses a $\mathcal{F}$ to a latent representation $z$. Then, we take latent representations $z$ to train our diffusion model. Thus, by extracting the compression process from the generative learning phase, we can speed up the diffusion model training process and reduce the memory footprints. And since the latent space is perceptually equivalent to the input space, there is no quality reduction for the diffusion model. We provide detailed descriptions of VAE and the diffusion model below.

*VAE.* A VAE comprises an encoder $E$ and a decoder $D$. Given a face tensor $\mathcal{F} \in \mathbb{B}^{|\mathcal{E}|}$, we first normalize $\mathcal{F}$ to $[-1.0, 1.0]$ using min-max normalization. Then $E$ encodes $\mathcal{F}$ to a mean code $\mu \in \mathbb{R}^{c \times d \times h \times w}$ and a standard deviation code $\sigma \in \mathbb{R}^{c \times d \times h \times w}$. We use the mean code $\mu$ as the latent code $z \in \mathbb{R}^{c \times d \times h \times w}$ without reparameterization, which differs from the typical VAE. Finally, $D$ decodes $z$ back to the face tensor $\mathcal{F}_{pred} = D(z)$. We train our VAE with the mean squared error (MSE) loss and the Kullback–Leibler divergence (KL) loss:

$$\mathcal{L}_{vae} = \mathcal{L}_{mse}(D(E(\mathcal{F})), \mathcal{F}) + KL(\mathcal{N}(\mu, \sigma) || \mathcal{N}(0, 1)). \quad (6)$$

Since we do not use the reparameterization technique, our VAE is more like an autoencoder (AE) producing compact latent codes (close to zero).

*Diffusion Model.* After encoding the face part $\mathcal{F}$ to the latent code $z$ with our VAE, we employ a diffusion model [24, 55] to the latent code distribution $p(z)$. We first normalize $z_0 \in p(z)$ to $[-1.0, 1.0]$. Then, through a series of diffusion steps, we introduce the controlled Gaussian noise $\epsilon$ to $z_0$ and transform it to $z_t = \sqrt{\bar{\alpha}_t} z_0 + \sqrt{1 - \bar{\alpha}_t} \epsilon$, where $t = 1 \dots T$ and $\bar{\alpha}_t = \prod_{i=1}^{t} \alpha_i$. $\alpha_t = 1 - \beta_t$ and $\beta_t$ is the predefined variance. The diffusion model $\theta$ is trained to predict the noise $\epsilon$, aiming at reversing the diffusion steps. The training objective is

$$\mathcal{L}_{dm} = \mathbb{E}_{x,t,\epsilon \sim \mathcal{N}(0,1)} ||\epsilon - \epsilon_{\theta}(z_t, t)||_1. \quad (7)$$

After training, to generate a face part $\mathcal{F}$, a sampled Gaussian noise $\epsilon \sim \mathcal{N}(0, 1)$ is seen as $z_T$. Then, our trained diffusion model denoises $z_T$ to $z_0$. $z_0$ is further denormalized from $[-1.0, 1.0]$ to the original

data range of $p(z)$. The VAE decoder takes denormalized $z_0$ as the input and decodes it to $\mathcal{F}$.

More details of the network are in the supplemental material.

### 3.4 Vertex Part Generation

The vertex part $\mathcal{V}$ is a set of relative vertex coordinates, and all vertices are arranged in a regular grid. Each vertex is in a cube of this grid. The vertex part contains all actual and potential vertices of a mesh. Since several vertices compose a face, there is a tight correlation between $\mathcal{V}$ and $\mathcal{F}$. Therefore, learning this correlation is a crucial problem in the vertex part generation.

In UDC, when the face part is determined, the rough shape is known, and the variance of the vertex part is limited. So we treat the vertex part generation as a regression task. We take a vertex refiner to generate $\mathcal{V}$ conditioned on $\mathcal{F}$. Here, we use a 3D version of U-Net [56] as the vertex refiner. Note that $\mathcal{F}$ is padded to the same size as $\mathcal{V}$ described in Sec. 3.2. Firstly, we normalize the face part $\mathcal{F}$ and the vertex part $\mathcal{V}$ to $[-1.0, 1.0]$. Secondly, the vertex refiner takes the $\mathcal{F}$ as the conditional input and generates a vertex part $\mathcal{V}_{pred} \in \mathbb{R}^{3 \times |C|}$ as shown in Fig. 2. We compare $\mathcal{V}_{pred}$ with the ground truth $\mathcal{V}_{gt}$ to train networks:

$$\mathcal{L}_{float} = \mathcal{L}_{mse}(\mathcal{V}_{gt}, \mathcal{V}_{pred}), \qquad (8)$$
$$\mathcal{V}_{pred} = \text{Unet3D}(\mathcal{F}), \qquad (9)$$

where $\mathcal{L}_{mse}$ is MSE loss. $\mathcal{V}_{gt}$ is the ground truth vertices paired with $\mathcal{F}$. In the inference phase, $\mathcal{V}_{pred}$ is denormalized from $[-1.0, 1.0]$ to $[0.0, 1.0]$.

This is a natural and efficient solution to learn the correlation between $\mathcal{F}$ and $\mathcal{V}$ for vertex part generation with reasonable training costs. If we eliminate the vertex refiner and generate $\mathcal{F}$ and $\mathcal{V}$ together, synthesized meshes will contain jagged edges due to inaccurate vertex coordinates. We will illustrate the necessity of the vertex refiner in Sec. 4.4.

More details of the network are in the supplemental material.

### 3.5 Implementation Details

If not specified otherwise, we set $X = Y = Z = 64$ for the grid $\mathcal{G}$ and $c = 64$, $d = h = w = 16$ for the latent code $z$. During training, $\mathcal{V}$ and $\mathcal{F}$ are normalized to $[-1.0, 1.0]$. At the final step of mesh generation, we denormalize the generated $\mathcal{V}$ and $\mathcal{F}$ to $[0.0, 1.0]$ and keep $\mathcal{V}$ as floating-point numbers and $\mathcal{F}$ as boolean values. We train the VAE and U-Net with all five categories as told in Sec. 4.1. In contrast, the diffusion model is trained in a category-specific manner. We use the AdamW optimizer [44] with $\beta_1$, $\beta_2 = [0.9, 0.999]$ for all networks. Empirically, large $\beta$ values can make our diffusion model produce realistic meshes. During the inference of diffusion models, we adopt the sampling method in Denoising Diffusion Probabilistic Models [24] and set the inference step as 1000.

## 4 EXPERIMENTS

### 4.1 Data

Following the protocol of MeshDiffuision [42], we use the ShapeNet Core (v1) dataset [5] to train and test our networks. Airplane, car, chair, refile, and table — five categories are used in our experiments. For each category, we split all data like [23] and [77] do: 70% as the training set, 20% as the test set, 10% as the validation set. To

be clear, the validation set is not used. For a fair comparison, we remove the interior of shapes. We apply the data-fitting method in Sec. 3.2 to generate UDC for all mesh data.

### 4.2 Shape Generation

To evaluate the quality of shape generation, we compare our method with IM-GAN [12], SDF-StyleGAN [77], MeshDiffusion [42] and, LAS-Diffusion [78]. IM-GAN, SDF-StyleGAN, and LAS-Diffusion are neural implicit function-based shape generation methods. IM-GAN predicts the occupancy values. Similarly, SDF-StyleGAN and LAS-Diffusion predict the SDFs. We apply MC to create meshes from synthesized implicit representations, following their protocols. MeshDiffusion is also a mesh generation method that utilizes a deformable tetrahedral grid and SDF values to generate meshes directly. We do not compare ours with PolyGen, MeshGPT, and PolyDiff because it is unfair that their faces are limited to no more than 2800, and they cannot produce complex geometric shapes.

Four metrics and three kinds of distances are used in the quantitative experiments as shown in Tab. 2. We take the test dataset as the reference set $\mathcal{B}$ and generate samples $\mathcal{A}$ of the same number, i.e. $|\mathcal{A}| = |\mathcal{B}|$. To calculate chamfer distance (COV) and earth mover's distance (EMD), we sample 2048 points for each mesh of $\mathcal{A}$ and $\mathcal{B}$. Note that all point clouds are normalized to [-1.0, 1.0], and meshes are normalized to [-0.5, 0.5]. More details of metrics are elaborated in the supplementary materials.

*Quantitative evaluation.* We present metric values in Tab. 2. Our method outperforms others in most cases, indicating that our approach is superior in terms of quality, diversity, and distribution. Particularly in the car and airplane category, our method demonstrates significantly better performance than others. It can be attributed to our excellent ability to generate details, considering the minimal intra-class variation within cars. We also achieve good performance in high resolution, shown in Tab. 3.

*Qualitative evaluation.* We show rendered meshes of various methods in Fig. 3. As seen, neural implicit function-based methods tend to produce over-smooth shapes and inaccurate parts, e.g., arms of chairs, wheels of cars, and legs of tables. MeshDiffusion usually produces pits on surfaces due to the ambiguity and inaccurate 2D supervision, which we have examined the reason in Sec. 2.1. The Laplacian smoothing used in MeshDiffusion reduces its generation quality by removing details and thin parts, such as chair arms and legs, rifle barrels, aero engines, and airplane propellers. In contrast, our GenUDC can generate high-quality meshes with realistic appearances, various structures, and rich details. We provide more visual samples in the supplementary materials.

### 4.3 Comparison with NDC

Since NDC [10] is an isosurface reconstruction method, we cannot directly compare GenUDC with NDC. Thus, we train NDC and UNDC networks with the default setting of their codes and the data from the airplane category of ShapeNetCore (v1) [5] for 2500 epochs. After training, we apply them to SDFs to create meshes for comparison. Qualitative evaluations are shown in Tab. 4. The performance of NDC and UDNC is poor due to the distribution gap between SDFs generated by SDF-StyleGAN and SDFs for training

**Table 2: Quantitative evaluation of shape generation in $64^3$ resolution.**

| | Method | MMD (↓) | | | COV (%, ↑) | | | 1-NNA (%, ↓) | | | JSD ($10^{-3}$, ↓) |
|---|---|---|---|---|---|---|---|---|---|---|---|
| | | CD $\times10^3$ | EMD $\times10$ | LFD | CD | EMD | LFD | CD | EMD | LFD | |
| Chair | IM-GAN | 13.928 | 1.816 | 3615 | **49.64** | 41.96 | 47.79 | 58.59 | 69.05 | 68.58 | 6.298 |
| | SDF-StyleGAN | 15.763 | 1.839 | 3730 | 45.60 | 45.50 | 43.95 | 63.25 | 67.80 | 67.66 | 6.846 |
| | MeshDiffusion | **13.212** | 1.731 | 3472 | 46.00 | 46.71 | 42.11 | **53.69** | **57.63** | 63.02 | 5.038 |
| | **Ours** | 14.083 | **1.653** | 2924 | 48.08 | **48.60** | 47.94 | 59.18 | 58.67 | 60.84 | **4.837** |
| Car | IM-GAN | 5.209 | 1.197 | 2645 | 28.26 | 24.92 | 30.73 | 95.69 | 94.79 | 89.30 | 42.586 |
| | SDF-StyleGAN | 5.064 | 1.152 | 2623 | 29.93 | 32.06 | 41.93 | 88.34 | 88.31 | 84.13 | 15.960 |
| | MeshDiffusion | 4.972 | 1.196 | 2477 | 34.07 | 25.85 | 37.53 | 81.43 | 87.84 | 70.83 | 12.384 |
| | **Ours** | **3.753** | **0.854** | 1191 | **45.67** | **46.53** | **45.73** | **60.80** | 58.33 | 62.23 | **2.839** |
| Airplane | IM-GAN | 3.736 | 1.110 | 4939 | 44.25 | 37.08 | **45.86** | 79.48 | 82.94 | 79.11 | 21.151 |
| | SDF-StyleGAN | 4.558 | 1.180 | 5326 | 40.67 | 32.63 | 38.20 | 85.48 | 87.08 | 84.73 | 26.304 |
| | MeshDiffusion | **3.612** | 1.042 | 4538 | 47.34 | 42.15 | 45.36 | 66.44 | 76.26 | **67.24** | 11.366 |
| | **Ours** | 3.960 | **0.902** | 3167 | 48.33 | 50.06 | 44.13 | 60.75 | 56.74 | 69.16 | **7.020** |
| Rifle | IM-GAN | 3.550 | 1.058 | 6240 | 46.53 | 37.89 | 42.32 | 70.00 | 72.74 | 69.26 | 25.704 |
| | SDF-StyleGAN | 4.100 | 1.069 | 6475 | 46.53 | 40.21 | 41.47 | 73.68 | 73.16 | 76.84 | 33.624 |
| | MeshDiffusion | **3.124** | 1.018 | 5951 | **52.63** | 42.11 | 48.84 | 57.68 | 67.79 | **55.58** | 19.353 |
| | **Ours** | 3.530 | **0.849** | 3493 | 48.42 | 51.58 | 50.53 | 56.63 | 55.05 | 55.58 | **10.951** |
| Table | IM-GAN | **11.378** | 1.567 | 3400 | **51.04** | 49.20 | 51.04 | 65.96 | 63.17 | 62.49 | 4.865 |
| | SDF-StyleGAN | 13.896 | 1.615 | 3423 | 42.21 | 41.80 | 42.98 | 68.35 | 68.21 | 66.19 | 4.603 |
| | MeshDiffusion | 11.405 | **1.548** | 3427 | 49.56 | 50.33 | **51.92** | 59.35 | 59.47 | 58.97 | 4.310 |
| | **Ours** | 11.998 | 1.564 | 2683 | 46.36 | 50.41 | 47.12 | 61.46 | 59.43 | 60.75 | **3.822** |

**Table 3: Quantitative evaluation of shape generation in $128^3$ resolution on airplane category.**

| Method | MMD (↓) | | | COV (%, ↑) | | | 1-NNA (%, ↓) | | | JSD ($10^{-3}$, ↓) |
|---|---|---|---|---|---|---|---|---|---|---|
| | CD $\times10^3$ | EMD $\times10$ | LFD | CD | EMD | LFD | CD | EMD | LFD | |
| LAS-Diffusion | 4.654 | 0.56 | 3142 | 37.45 | 35.72 | 42.15 | 79.48 | 84.67 | 71.51 | 33.137 |
| **Ours** | **4.000** | **0.509** | 3077 | **46.72** | **43.88** | **42.27** | **60.01** | **61.06** | **69.22** | **6.873** |

**Table 4: Quantitative comparison between MC, NDC, UNDC, and ours in $64^3$ resolution on the airplane category. We apply those three methods to the same SDFs generated by SDF-StyleGAN. A post-processing step described in [10] is used after UNDC.**

| Method | MMD (↓) | | | COV (%, ↑) | | | 1-NNA (%, ↓) | | | JSD ($10^{-3}$, ↓) |
|---|---|---|---|---|---|---|---|---|---|---|
| | CD $\times10^3$ | EMD $\times10$ | LFD | CD | EMD | LFD | CD | EMD | LFD | |
| SDF-StyleGAN + MC | 4.459 | 1.113 | 3731 | 41.29 | 43.88 | 41.14 | 81.33 | 76.89 | 80.04 | 20.581 |
| SDF-StyleGAN + NDC | 7.341 | 1.257 | 3748 | 17.55 | 20.64 | 44.00 | 94.13 | 95.30 | 78.06 | 133.827 |
| SDF-StyleGAN + UNDC | 7.758 | 1.563 | 3902 | 15.57 | 14.46 | 41.90 | 95.24 | 97.71 | 80.66 | 173.030 |
| **Ours** | **3.960** | **0.902** | 3167 | **48.33** | **50.06** | **44.13** | **60.75** | **56.74** | **69.16** | **7.020** |

NDC, shown in Fig. 4. Fig. 5 shows that NDC and UDNC cannot handle the generated SDFs, resulting in surface distortion and floating artifacts. Overall, integrating NDC (UNDC) into the SDF generation method introduces too many uncertainties, making it unsuitable for mesh generation. In contrast, our GenUDC directly generates high-quality meshes using UDC, demonstrating that our paradigm is more suitable for mesh generation.

## 4.4 Ablation Study of the Vertex Part Generation

In this section, we compare the GenUDC to the one without the U-Net to demonstrate the necessity of the vertex refiner, i.e., U-Net.

In the one without the U-Net, we concatenate the face part $\mathcal{F}$ and vertex part $\mathcal{V}$ as a mesh tensor and then use the LDM to learn the distribution of mesh tensors. Other settings are consistent with the vanilla GenUDC. More network details are in the supplementary materials. Then, we take mesh tensors to train the LDM, learning the joint distribution of $\mathcal{F}$ and $\mathcal{V}$. At the inference, it simultaneously generates $\mathcal{F}$ and $\mathcal{V}$. However, it is quite difficult for a single LDM to learn this joint distribution and build the correlation between $\mathcal{F}$ and $\mathcal{V}$. To prove this, we present some similar samples produced by GenUDC with and without U-Net in Fig. 6. As we can see, removing U-Net results in jagged edges and unsmooth surfaces. Only by modeling the vertex part generation conditioned on the face part,

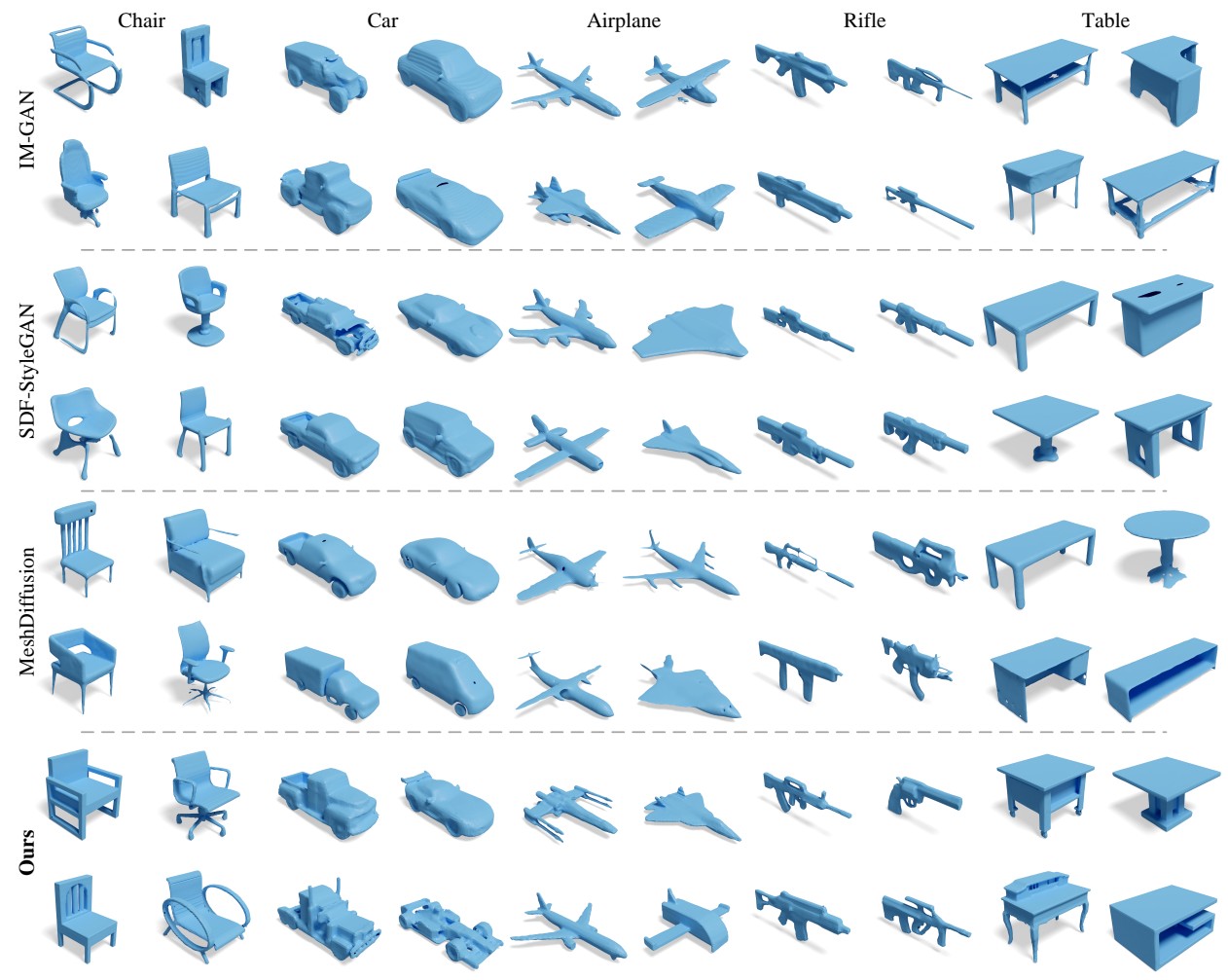

Figure 3: Qualitative evaluation of shape generation in $64^3$ resolution.

Table 5: Quantitative evaluation of ablation study. We compare two methods on the car category following the setting in Sec. 4.2.

| Method | MMD (↓) | | | COV (%, ↑) | | | 1-NNA (%, ↓) | | | JSD ($10^{-3}$, ↓) |
|---|---|---|---|---|---|---|---|---|---|---|
| | CD $\times 10^3$ | EMD $\times 10$ | LFD | CD | EMD | LFD | CD | EMD | LFD | |
| Ours w/o U-net | 15.463 | 1.702 | 3073 | 36.28 | 42.77 | 36.06 | 74.93 | 72.23 | 75.18 | 6.574 |
| Ours w/ U-net | **14.083** | **1.653** | **2924** | **48.08** | **48.60** | **47.94** | **59.18** | **58.67** | **60.84** | **4.837** |

we can learn the correlation between $\mathcal{F}$ and $\mathcal{V}$. The quantitative evaluation in Tab. 5 also proves our opinion.

## 4.5 Data Fitting Comparison

Our UDC is a discretized mesh counterpart, which requires a data fitting process. In this section, we demonstrate the superiority of UDC in the data fitting process compared with MeshDiffusion, which uses a deformable tetrahedral grid to discretize a mesh.

For quantitative evaluation, we randomly select one hundred meshes and record the average processing time and memory footprints in Tab. 6. As shown, UDC outperforms MeshDiffusion in both speed and memory footprint. The reason is that MeshDiffusion uses the rendered 2D images as the supervision of data fitting. Rendering 2D images requires a lot of GPU and CPU resources, and it takes a long time to fit data. In contrast, we only use the CPU to directly calculate the fitting vertices and faces of UDC as we elaborate in Sec. 3.2, which is resource-efficient and fast.

To visually illustrate UDC's superiority, we present some samples in Fig. 7. As seen, MeshDiffusion is unavoidable to produce pits on the mesh surfaces and lack of details, such as the line and crack on the car. The reason is the ambiguity and inaccurate 2D supervision talked about in Sec. 2.1. Laplacian smoothing used by

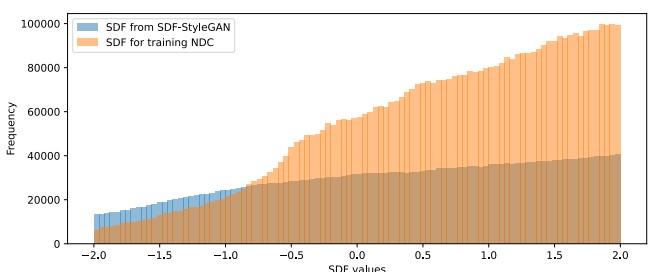

Figure 4: The histogram of SDFs generated by SDF-StyleGAN and SDFs for training NDC. We select 809 SDF grids and only consider SDFs near the surfaces to draw this histogram.

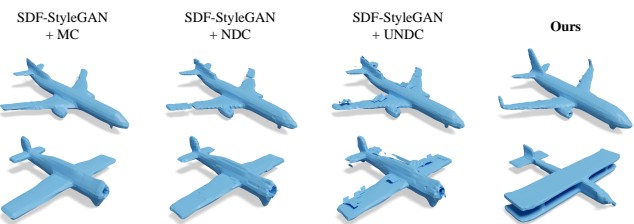

Figure 5: Visual samples of three post-processing methods and ours. We apply those post-processing methods on the same SDFs generated by SDF-StyleGAN.

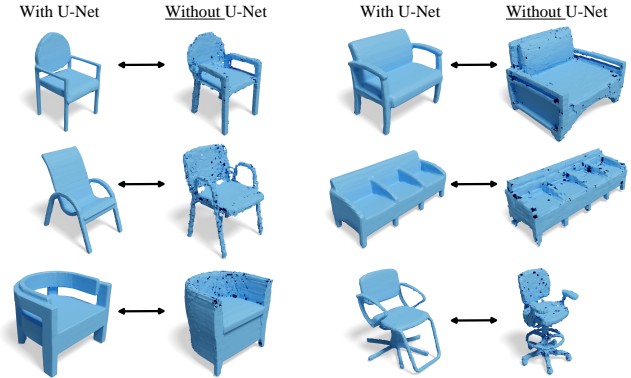

Figure 6: GenUDC with U-Net vs. GenUDC without U-Net. A pair of samples are not the same object but are similar in appearance and structure.

MeshDiffusion even removes details and sharp parts instead of the pits. In comparison, UDC can fit flat surfaces, sharp parts, and curved surfaces with details.

## 5 LIMITATION

Firstly, the main limitation of our method is the non-manifold issue. Since we adopt UDC as the mesh representation, our method inherits the non-manifold issue from DC. However, such an issue rarely occurs. It can be resolved by "tunneling" through vertices/edges or dividing them with the approaches introduced by [52, 58]. Secondly, the memory footprint constrains the resolution of our results.

Table 6: Quantitative evaluation of data fitting in terms of mean processing time, GPU memory footprints, and CPU memory footprints of one hundred samples. All the programs are executed in a single thread, using an NVIDIA RTX 3090 GPU, an Intel i7-10700 CPU, and 64GB of memory.

| | Processing time (Sec.) | GPU memory (MB) | CPU memory (MB) |
|---|---|---|---|
| MeshDiffusion | 1,408 | 8544 | 1497 |
| **Ours (UDC)** | **43** | **0** | **1266** |

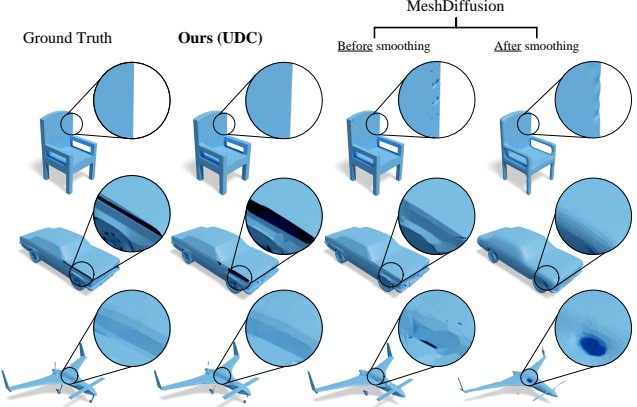

Figure 7: Qualitative evaluation of data fitting. The resolution of those meshes is $64^3$ except for the ground truth. In addition, since MeshDiffusion applies a post-processing method to the uneven surfaces of its fitting mesh, we present both raw and smoothed meshes.

Thirdly, since the face part is a set of boolean values, our models may predict wrong boolean values, resulting in pits on the surface. We can solve this pit problem with the post-processing method in [10].

## 6 CONCLUSION & FUTURE WORKS

In conclusion, we propose a novel 3D generative framework, GenUDC, using the Unsigned Dual Contouring representation (UDC) for high-quality mesh generation. Our method can directly generate high-quality meshes without using isosurface reconstruction methods. Specifically, following the discretization idea, we fit a mesh in a regular grid to get its UDC representation. Since UDC is composed of the face and vertex parts, we use a two-stage, coarse-to-fine pipeline to learn its distribution. Firstly, we use a latent diffusion model to generate the face part. Secondly, we take a U-Net as a vertex refiner to synthesize the vertex part conditioned on the face part. Experiments demonstrate our superiority over baselines in shape generation and data fitting. The ablation study proves the validity of network design. We believe that our method offers a new paradigm for further work in mesh generation.

In the future, we plan to apply GenUDC to various applications, such as text-to-3D, joint generation of texture and shape, single view 3D reconstruction, shape editing, etc.

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
