# OpenReview forum: "GenUDC: High Quality 3D Mesh Generation With Unsigned Dual Contouring Representation"
_acmmm.org/ACMMM/2024/Conference — MM2024 Poster_

### Official Review · Reviewer_AfTh · 2024-05-24

**Rating:** 5
**Confidence:** 2

**Summary:**

This paper proposes to employ Unsigned Dual Contouring (UDC) as the 3D representation for 3D mesh generation, which exhibits high-quality results. UDC representation converts mesh to a grid-based representation, which can be divided into two parts: vertex parts for expressing high-quality details and face parts for describing the overall shape. Based on this dividing, they propose a method using two-stage, coarse-to-fine generative process to synthesize high-quality results.

**Strengths:**

This paper leverages a novel representation of 3D mesh and produces high-quality results.
Exhaustive experiments validate the superiority of their methods.

**Limitations:**

Writing:

1.	In the first paragraph:“However, Employing” -> “However, employing”

2.	It’s unclear what are the two challenges in line 126. (Slow preparation? Crumpled? Limits the number of faces?)

3.	Carefully redraft Paragraph 2 to demonstrate the core challenge(s) of this task. 1) “No more than a specific number (2800)” without any explanation is nonsense. There must be a more fundamental reason. 2) Why does the deformable nature of the grid lead to crumpled mesh? Do you forget any citations or explanations there? 3) Why is 2D supervision there? It’s different from your input. Besides, it can be solved by providing accurate 2D supervision and hence it does not highlight the need of a new 3D representation.

4.	Do not repeat two-stage, coarse-to-fine twice in the same paragraph.


Overall I like the idea and the results exhibited in this paper. However, I have some concerns.
1.	If F are Boolean values, why is it necessary to normalize it using min-max?

2.	UDC representation is still limited by the grid resolution like other grid-based representations (e.g., SDF). What are the fundamental features that distinguish it from the SDF representation in terms of better quality?

3. Can it generalize to objects with different orientations?

4.	The synthesized vertices might be incompatible with the faces. How do you handle this situation?

5.	This paper mentions that text-to-3D would be a future application. Are all the results in the paper synthesized by unconditional generation? How do you generate results in the same category from a noise?

**Suitability:**

2

---

### Official Review · Reviewer_nfF4 · 2024-05-25

**Rating:** 2
**Confidence:** 3

**Summary:**

This article introduces a novel generative framework called GenUDC, which aims to generate high-quality 3D meshes with complex structures and realistic surfaces using Unsigned Dual Contouring (UDC) as the mesh representation. Compared to existing methods based on sequence data or deformable tetrahedral meshes, GenUDC addresses challenges in mesh generation through UDC, achieving precise reconstruction of both face and vertex parts and thus solving the ambiguity problem of deformable meshes. The framework employs a two-stage, coarse-to-fine generation process: first generating the face parts of the rough shape, followed by generating the detailed vertex parts. Experimental results demonstrate that GenUDC excels in mesh generation, being fast and memory-efficient while producing high-quality meshes with intricate structures and realistic details.

**Strengths:**

The GenUDC framework proposed in this article uses the UDC representation to address the challenges of generating complex structures and realistic surfaces, ensuring the creation of high-quality meshes. It employs a two-stage generation process, starting with a coarse stage followed by a fine stage, effectively preserving the accuracy and details of the mesh. GenUDC excels in speed and memory efficiency, capable of efficiently generating meshes with intricate structures and realistic details.

**Limitations:**

1.The article lacks an in-depth discussion and comparative analysis of the GenUDC framework. It is recommended to highlight its innovations in generating complex structures and realistic surfaces.
2.There are some deficiencies in the experimental design and result presentation. The article lacks a comprehensive comparison and in-depth analysis with other related methods, making it difficult for readers to accurately assess the practical application and advantages of the GenUDC framework in the field of mesh generation.
3.The article uses only one dataset. It is suggested to add two more datasets for comparative analysis. Besides analyzing the advantages of your model, please also analyze its disadvantages. The dataset used in the article may have biases or be insufficient to comprehensively evaluate the performance of the GenUDC framework.
4. There is a lack of discussion on the advantages and limitations of the method as a mesh representation.
5. It is recommended that the authors enhance the discussion on techniques such as neural rendering in future research, improve the experimental design and result presentation, and thereby enhance the academic quality and credibility of the article.

**Suitability:**

2

---

### Official Review · Reviewer_gNsr · 2024-05-25

**Rating:** 4
**Confidence:** 3

**Summary:**

The paper introduces the GenUDC framework for high-quality 3D mesh generation, GenUDC uses Unsigned Dual Contouring (UDC) to represent meshes, discretizing them into face and vertex components to accurately recover complex structures and fine details. This approach resolves ambiguity issues through a one-to-one mapping between UDC and the mesh. GenUDC employs a two-stage generative process: first generating a coarse shape via the face part, then refining it with detailed vertices.

**Strengths:**

1. The paper is clearly written and figures are useful.
2. Qualitative and quantitative evaluations and comparisons with baselines show that the method outperforms the current methods on mesh generation.
3. The first work that utilizing UDC as the representation for high-quality mesh generation.

**Limitations:**

1. The results in Table 2 are not good for the Chair and Table categories, but there are better results for other categories. Why is this? Is it because the UDC representation is not suitable for this type of geometric generation?
2. Why was there no comparison with Get3D[1]?


[1] Gao J, Shen T, Wang Z, et al. Get3d: A generative model of high quality 3d textured shapes learned from images[J]. Advances In Neural Information Processing Systems, 2022, 35: 31841-31854.

**Suitability:**

3

---

### Meta-Review · Area_Chair_XQNE · 2024-06-29

**Recommendation:** Accept (Poster)
**Confidence:** 5

**Metareview:**

The paper proposes a generative framework, which aims to generate high-quality 3D meshes with complex structures and realistic surfaces using UDC as the mesh representation. The proposed method addresses challenges in mesh generation via UDC.

This paper accept two final borderline accepts and one initial weak reject. The reviewer who rates as weak reject didn't update the final recommendation. I read the author's rebuttal, and found that the rebuttal well address the second reviewer's concern. Therefore the paper can be accepted for publication at ACM MM.